# Histone H3 N-Terminal Tail Residues Important for Meiosis in *Saccharomyces cerevisiae*

**DOI:** 10.3390/biom15081202

**Published:** 2025-08-21

**Authors:** Amy Prichard, Marnie Johansson, David T. Kirkpatrick, Duncan J. Clarke

**Affiliations:** Department of Genetics, Cell Biology & Development, University of Minnesota, Minneapolis, MN 55455, USA

**Keywords:** histone H3, meiosis, phosphorylation, H3T3, sporulation, histone modifications

## Abstract

Histone tail phosphorylation has diverse effects on a myriad of cellular processes, including cell division, and is highly conserved throughout eukaryotes. Histone H3 phosphorylation at threonine 3 (H3T3) during mitosis occurs at the inner centromeres and is required for proper biorientation of chromosomes on the mitotic spindle. While H3T3 is also phosphorylated during meiosis, a possible role for this modification has not been tested. Here, we asked if H3T3 phosphorylation is important for meiotic division by quantifying sporulation efficiency and spore viability in *Saccharomyces cerevisiae* mutants with a T3A amino acid substitution. The T3A substitution resulted in reduced sporulation efficiency and reduced spore viability. Analysis of two other H3 tail mutants, K4A and S10A, revealed different effects on sporulation efficiency and spore viability compared to the T3A mutant, suggesting that these phenotypes may be due to failures in distinct functions. To determine if the spindle checkpoint promotes spore viability of the T3A mutant, the *MAD2* gene was deleted. This resulted in a severe reduction in spore viability following meiosis. Altogether, the data reveal an important function for histone H3 threonine 3 that requires monitoring by the spindle checkpoint to ensure successful completion of meiosis.

## 1. Introduction

Meiosis generates haploid progeny from diploid cells after two rounds of chromosome segregation, and its mechanisms are highly conserved throughout eukaryotes, from yeast to humans [1]. In meiosis I, cells segregate duplicated homologous chromosomes, and in meiosis II, the sister chromatids are segregated. In *Saccharomyces cerevisiae*, meiosis is accompanied by sporulation, which reorganizes the mother cell cytoplasm to produce an ascus in which individual plasma membranes and cell walls encapsulate the four haploid cells (spores) [2]. The visually perspicuous nature of sporulation provides the opportunity to examine the ability of cells to complete meiosis successfully, accurately segregating the chromosomes and producing mature asci.

Histones are the central proteins that make up nucleosomes. Each nucleosome is an octamer that contains two heterodimers of histone H2A and histone H2B and a heterotetramer of two histone H3 and two histone H4 proteins. Histones are globular proteins with largely unstructured N-terminal tails. The tails are approximately 25–35 amino acids in length with a high proportion of basic residues. Both the N-terminal tails and the globular histone core domains are subject to post-translational modifications [3]. Commonly, histones can be phosphorylated, methylated, and acetylated on their N-terminal tails. A conserved feature of meiosis in eukaryotes is that it is accompanied by such histone post-translational modifications, and accumulating evidence indicates that these modified residues orchestrate meiosis-specific gene expression patterns, chromosomal structural changes and recombination, and kinetochore–microtubule attachment ahead of meiotic chromosome segregation [4,5,6,7,8,9]. Indeed, histone modifications can have numerous effects, such as changing chromatin structure and recruiting other proteins to the chromatin, which in turn controls cellular processes including transcription, DNA repair, and cell division [10,11]. Several specific modified residues have been shown to be important for meiosis in yeast. For example, histone H3 threonine 11 is phosphorylated by the meiosis-specific kinase Mek1 in response to double-stranded DNA breaks (DSBs) in meiosis and plays a role in meiotic recombination [12]. Histone H3 lysine 79 methylation promotes DSB formation in meiosis, probably acting redundantly with histone H3 lysine 4 methylation [13]. Histone H4 lysine 16 acetylation is important for cells to activate meiotic checkpoints when synaptonemal complex defects arise [14].

In mitosis, a prominent histone H3 modification is phosphorylation at threonine 3 (T3) by Haspin kinase [15]. This modification primarily occurs on histone H3 proteins at the inner centromeres of chromosomes [16,17,18,19], where it serves to recruit the Chromosome Passenger Complex (CPC) including Aurora B kinase [18,20,21,22,23]. Given the essential roles of Aurora B at centromeres in mitosis, Haspin-mediated T3 phosphorylation is needed for proper chromosome alignment on the spindle and accurate chromosome segregation [16,17]. In addition, H3T3 phosphorylation has been implicated in chromatin compaction in yeast [24].

H3T3 is also phosphorylated at centromeres during meiosis I and II in mouse oocytes [25]. Although a possible requirement for T3 phosphorylation in meiosis has not been directly tested, inhibition of Haspin kinase causes defects in kinetochore–microtubule attachment, and aneuploidy is observed following the completion of meiosis [25,26]. Recent studies also detected H3T3 phosphorylation in mitotic *S. cerevisiae* cells [22]. Proper localization of the yeast ortholog of Aurora B kinase (Ipl1) was found to be disrupted when T3 phosphorylation was abrogated during mitosis [22]. However, whether H3T3 has a role in meiosis in yeast has not been fully evaluated.

Yeast serves as a powerful model to study meiosis, and previous work identified histone H3 and H4 residues required for yeast sporulation [27]. Here we used *Saccharomyces cerevisiae* W303 as a model to determine whether certain histone tail residues, including H3T3, are important for meiotic division and whether this effect is dependent on Mad2-dependent spindle assembly checkpoint function.

## 2. Materials and Methods

Yeast strains: The strains used were all derived from *Saccharomyces cerevisiae* strain W303, as listed in Table 1. The histone H3 mutants are at the primary H3 locus, *HHT2*, in strains with the secondary H3 and H4 loci, *hht1-hhf1*, deleted [28].

Yeast media: Three types of growth media were used in this study: YPD (Yeast Extract, Peptone, Dextrose) medium, SD (Synthetic Defined) drop-out medium, and sporulation medium. YPD liquid medium consisted of 10 g/L Bacto-Yeast Extract, 20 g/L Bacto-Peptone, and 20 g/L Dextrose, and YPD solid medium consisted of the same ingredients with the addition of 20 g/L agar. SD drop-out medium consisted of 10 g/L potassium acetate, 1 g/L Yeast Extract, 0.5 g/L Dextrose, 20 g/L agar, 6 mg/L adenine, and 1.4 g/L drop-out mix (1 g l-adenine, 1 g l-uracil, 2 g l-tryptophan, 1 g l-histidine, 1 g l-arginine, 1 g l-methionine, 3 g l-tyrosine, 4 g l-leucine, 4 g l-isoleucine, 3 g l-lysine, 2.5 g l-phenylalanine, 5 g l-glutamate, 5 g l-aspartate, 7.5 g l-valine, 10 g l-threonine, and 20 g l-serine), with 0.5 mL 4 M NaOH added per liter of medium. Sporulation medium consisted of 10 g/L potassium acetate, 1 g/L Yeast Extract, 0.5 g/L Dextrose, 20 g/L agar, and 6 mg/L adenine.

Yeast strain mating: Forced mating was employed to generate crosses of different yeast strains. To achieve this, two yeast cells of opposite mating types were placed next to each other on solid YPD medium using a dissection microscope. After two days of growth at 30 °C, putative diploids were examined for the ability to produce mating pheromone to suppress haploid cell growth in halo assay tests.

Mating-type (halo) assay: To determine mating type of non-diploid colonies, tester strains that arrest cell growth in the presence of the opposite mating factor were plated on solid YPD medium. Small amounts of the yeast isolate being tested were transferred to the test plate and grown at 30 °C for two days. If the yeast isolate was of the opposite mating type to the tester, a halo where no tester cells could grow formed around the patch of yeast isolate. Diploid colonies do not form a halo on either the *a* or *α* test plates since they do not secrete mating factors.

Diploid yeast sporulation: Yeast cells were grown in 5 mL liquid YPD medium overnight for 16 h at 30 °C in a rotator. The cells were then centrifuged at 2000 rpm for 6 min to pellet in a tabletop centrifuge, washed three times with ddH_2_O, and plated on solid sporulation medium in 200 μL of ddH_2_O at room temperature. The cells were grown for three days at room temperature on sporulation medium before sporulation analysis and tetrad dissection.

Sporulation analysis: Sporulated cells were scraped from the solid sporulation medium, suspended in 5 μL ddH_2_O on a glass slide, and counted using a light microscope (Zeiss, Jena, Germany). The sporulation efficiency was calculated by categorizing each cell as either an ascus or non-ascus. Asci were defined as any cell that had attempted sporulation. These included cells that had completed the process of sporulation and had formed a mature ascus with either two, three, or four spores inside the ascus. Also included were cells that had not completed ascus maturation but had either two, three, or four spores within the round cell wall. Within each group, we also quantified successful sporulation (Table 2). Successful sporulation was defined as the proportion of the asci that were tetrads with complete ascus maturation, i.e., as opposed to mature two-, three-, or four-spore asci or immature asci that had begun but not completed sporulation.

Tetrad analysis: Sporulated cells were incubated with 1 mg/mL lyticase for 12 min at 37 °C before being plated on YPD medium. Tetrads were then dissected using a dissection microscope and grown for three days at 30 °C. The resulting individual colonies were counted to determine spore viability.

Statistical analysis: All statistics were analyzed using a χ^2^ test. This test was chosen since the data collected were discrete whole numbers: numbers of spores and numbers of viable tetrads. All experiments were repeated at least three times.

## 3. Results

### 3.1. Sporulation Efficiency of Yeast Cells Lacking Histone H3 Threonine 3

To evaluate possible requirements of H3T3 in meiosis, diploid yeast strains were made by crossing two wild-type strains to generate a wild-type homozygous diploid (DCY4706), crossing a wild-type and a T3A mutant to generate heterozygous T3A mutants (DCY4707 and DCY4824), and crossing two T3A strains to generate a homozygous T3A mutant (DCY4730) (Table 1). For the histone mutant strains, the histone H3 mutant alleles are integrated at the primary H3 locus, *HHT2*, in strains with the secondary H3 and H4 loci, *HHT1-HHF1*, deleted [28]. Each of the diploids were plated on sporulation medium, and the sporulation efficiency was quantified by light microscopy after three days (Figure 1). Sporulation efficiency was 38.9% in the wild-type homozygotes but was drastically reduced to only 1.9% in the homozygous T3A mutant. The two different heterozygous strains for the T3A substitution had sporulation efficiencies of 38.8% and 37.7%, very similar to the wild-type. The low sporulation frequency of the T3A homozygote reveals that the lack of H3T3 phosphorylation might negatively impact the ability of cells to initiate or complete meiosis. However, the sporulation process can clearly tolerate having only one-half of the histones with a phosphorylatable H3T3.

Next, we asked if the fidelity of sporulation was impacted in the T3A mutant (Table 2). Cells that attempted sporulation were categorized as having successfully completed sporulation if a mature four-spore ascus was produced (tetrad). Unsuccessful sporulation was defined as two-spore or three-spore asci that had completed sporulation or cells that had immature asci containing pro-spores (presumably delayed or arrested part-way through meiosis). Based on these categories, out of 1433 wild-type cells attempting sporulation, 86% had completed sporulation successfully. The heterozygotes completed sporulation successfully 80% and 78% of the time. Of 48 homozygous T3A cells attempting sporulation, only 44% completed sporulation successfully. Therefore, not only was sporulation efficiency reduced in the T3A mutant but successful completion of sporulation was perturbed.

### 3.2. Spore Viability in Yeast Cells Lacking H3T3

Sporulation efficiency of the T3A homozygote was only 1.9%, and of those, only 44% produced mature tetrads. This indicates a critical role for this histone H3 residue in meiosis. We next asked if mature tetrads produced by T3A mutant cells were viable. Tetrads were dissected to separate each spore and allow them to grow into individual colonies. We dissected at least 100 tetrads for each strain and then quantified the number of spores that grew into viable colonies (Figure 1). This revealed 94.5% of wild-type spores were viable, while only 59.3% of the homozygous T3A spores were viable. The viability of the heterozygotes was similar to the wild-type, at 90.3% and 95.5%. We then quantified the frequency with which tetrads had all four spores giving rise to viable colonies, i.e., the percentage of viable tetrads. Tetrad viability was 87.5% for wild-type and 81% and 90% for the heterozygotes but only 27.5% for the T3A mutant. Altogether, the T3A mutant had greatly reduced sporulation efficiency, a reduction in the frequency of sporulation resulting in the formation of mature four-spore asci, and significantly lower viability of the spores in those mature tetrads.

### 3.3. H3 N-Terminal Tail Mutations Differentially Affect Spore Viability

We next investigated if the effects the T3A mutation had on meiosis are specific to the threonine 3 residue or if mutation of other histone H3 tail residues have the same consequences. Homozygous and heterozygous histone H3 K4A and S10A mutants were generated, and their sporulation efficiency and spore viability were quantified as for the T3A mutants. We chose lysine 4 because it is directly adjacent to threonine 3, but rather than being phosphorylated, it can be methylated or acetylated. K4 methylation has been extensively studied and is known to be involved in regulation of gene expression pathways initiated during meiosis [29,30,31,32,33] as well as being associated with sites of meiotic recombination initiation in yeast [34,35]. We chose S10 because, although it is farther away from T3, it is also subject to phosphorylation and plays known roles in chromosome compaction during both mitosis and meiosis [36,37,38,39,40,41].

Compared to the wild-type, the S10A mutant had reduced sporulation efficiency (10.1%) and reduced spore viability (75.0%) (Figure 2), but these effects on meiosis were not as severe as the T3A mutant. The frequency of successful sporulation was also reduced in the S10A mutant (Table 2). The K4A mutant had the lowest sporulation efficiency, 0.4% (Figure 3). It had such a low sporulation efficiency that tetrad dissection and analysis were impractical. These differences between T3A, S10A, and K4A show that each mutation has a different effect on meiosis (statistical analysis is summarized in Figure 4). However, it is notable that all the mutations tested had deficits in sporulation efficiency. Overall, the results suggest that the H3T3 residue has a distinct contribution to meiosis compared to the S10 phosphorylation and K4 methylation/acetylation.

### 3.4. The Spindle Assembly Checkpoint Rescues Spore Viability in Meiosis in the Absence of H3T3 Phosphorylation

There were dramatic phenotypes in the T3A mutant: reduced sporulation efficiency and formation of aberrant asci in cells that attempted sporulation. The T3A mutant also had reduced spore viability in complete tetrads. We reasoned that these phenotypes could be associated with activation of the spindle assembly checkpoint (SAC), because in mitosis, H3T3 phosphorylation is required to promote proper microtubule–kinetochore interactions. The low sporulation efficiency and low spore viability in the T3A mutant might be due to improper kinetochore attachments, in which case the SAC might be triggered. The Mad2 protein is known to be required for the SAC in both mitosis and meiosis [42]. Therefore, we tested if sporulation efficiency and spore viability are compromised in T3A mutant cells lacking Mad2. A *mad2Δ* deletion mutation was crossed into the wild-type and the T3A mutant. We then quantified sporulation efficiency and spore viability for *mad2Δ* homozygotes and *mad2Δ* T3A homozygotes (Figure 5). The sporulation efficiency of the *mad2Δ* mutant was 35.7%, which is not significantly different from the wild-type (38.9%). The sporulation efficiency of the *mad2Δ* T3A double mutant was 2.4%, which is not significantly different from the T3A mutant (1.9%). Therefore, it does not seem that the SAC is restraining T3A mutants from progressing into or through the early stages of meiosis.

The spore viability of the *mad2Δ* mutant was 72.0% compared to 94.5% in wild-type cells. This is consistent with the meiotic spindle checkpoint protecting cells from spore inviability. The spore viability of the *mad2Δ* T3A double mutant was only 24.3%, compared with 59.3% in the T3A mutant. By multiplying the spore viability of the *mad2Δ* mutant (72%) by the spore viability of the T3A mutant (59.3%), we calculate the product of spore viabilities in the *mad2Δ* and T3A mutants to be 42.7%. This gives us an expected viability of 42.7% for the *mad2Δ* T3A double mutant. However, the spore viability of the *mad2Δ* T3A double mutant is in actuality 24.3%. This is significantly lower than our calculated expected value (χ^2^ test, *p*-value = 4.546 × 10^−^^8^), which assumes Mad2 and H3T3 have unrelated functions in meiosis. This suggests that the SAC is involved in partially rescuing the meiotic defects in T3A cells.

### 3.5. Spore Viability Patterns Suggest the Spindle Assembly Checkpoint Rescues Meiotic Chromosome Segregation Errors in the Absence of H3 T3 Phosphorylation

The viability of spores in H3 T3A mutants partly depended on the spindle assembly checkpoint that prevents chromosome mis-segregation. Inviability of haploid yeast can be due to chromosome segregation errors because chromosome loss in haploid cells is lethal. The pattern of spore viability after tetrad dissection can reveal chromosome segregation errors in meiosis I versus meiosis II. Mis-segregation of homologous chromosomes in meiosis I results in two viable spores, or no viable spores if more than one chromosome mis-segregates. In contrast, mis-segregation of a sister chromatid in meiosis II results in three viable spores and one inviable spore. Analysis of spore viability patterns in H3 T3A mutants revealed that the largest increase relative to the wild-type was in the two-viable-spores category (χ^2^ test, *p*-value = 9.502 × 10^−13^) (Figure 6). This was also the case for *mad2Δ* mutants (χ^2^ test, *p*-value = 1.057 × 10^−10^). These patterns are most consistent with meiosis I single homolog segregation errors. In the T3A *mad2Δ* double mutant, zero-spore viability was most frequent (χ^2^ test, *p*-value = 2.2 × 10^−16^), consistent with mis-segregation of multiple homologs in meiosis I.

## 4. Discussion

Phosphorylation of histone H3 at residue threonine 3 was detected in mitotic yeast cells and shown to be important for recruitment of Ipl1/Aurora B kinase to centromeres, as is the case in higher eukaryotes [16,17,18,22]. Whether H3 T3 phosphorylation is important in meiosis has not been directly studied. Here, we investigated a possible requirement for H3T3 phosphorylation during meiosis and found that this residue plays an important role in meiotic progression and fidelity. We observed that T3A homozygous mutants had decreased sporulation efficiency, and cells that did attempt sporulation often produced aberrant asci or were delayed or arrested as asci with immature pro-spores. In cells that produced mature tetrads, spore viability relative to the wild-type was significantly reduced. T3A heterozygotes behaved like wild-type cells, implying that these are recessive phenotypes. Interestingly, inhibition of Haspin kinase (which targets H3 T3) in mice perturbed meiosis in similar ways to the T3A mutation in yeast. In mice, Haspin inhibition delayed the onset of meiosis I by delaying progression from prophase I at the step of nuclear envelope breakdown [25,26]. Progression into anaphase I was also delayed, consistent with observed defects in kinetochore–microtubule attachment [25]. Haspin inhibition also led to aneuploidy after meiosis in the mouse oocytes. This is consistent with our finding that yeast T3A mutants had reduced spore viability in mature tetrads.

We also investigated if the effects of the T3A substitution were specific to that mutation or if altering other residues within the histone H3 N-terminal tail have the same consequences. By assaying sporulation efficiency and spore viability in histone H3 K4A and S10A mutants, we determined that the effects on meiosis are specific to T3A and are not general to other H3 tail mutations. Specifically, we observed that the T3A mutant has a higher sporulation efficiency than the K4A mutant, but a lower sporulation efficiency than the S10A mutant (Figure 4). We also observed that the T3A mutant has a lower spore viability than the S10A mutant (Figure 1 and Figure 2). Therefore, there are significant differences between the effects of the T3A, K4A, and S10A mutations in meiosis. The data could be explained by either of two possibilities. There could be mechanistic differences between the roles of these residues. Alternatively, the residues could share a common function yet have differential impacts for that shared role. For example, there could be some crosstalk between H3 T3 and H3 K4 in meiosis because it is reported that T3 phosphorylation abrogates the interaction of the meiosis protein Spp1 with methylated H3 K4 [43].

It is important to note that a previous study did not observe decreased sporulation efficiency in yeast T3A, K4A, or S10A strains [27]. In that case, a different strain background (SK1) was employed that has very high wild-type sporulation efficiency. One explanation for these apparent differences is that the strains used here (W303) have a reduced wild-type sporulation efficiency, which likely results in a sensitized genetic background allowing additional requirements for meiosis to be revealed.

Existing evidence suggests what roles histone H3 S10 phosphorylation and K4 methylation/acetylation play in meiosis versus T3 phosphorylation. S10 is phosphorylated in meiosis and is associated with chromosome compaction [36,44]. K4 is a known regulator of gene expression and may contribute to the meiosis-specific program of gene expression needed to induce sporulation and assure the fidelity of meiosis [45,46,47,48]. On the other hand, the most clearly defined roles of H3 T3 are to promote biorientation of chromosomes on the spindle and to delay the onset of anaphase when chromosomes cannot be segregated accurately [22,23,49,50,51,52,53,54,55,56,57]. To test if this also occurs in meiosis, it will be important to determine exactly where on chromosomes H3T3 is phosphorylated in meiotic cells. We predict H3 will be phosphorylated on T3 at centromeres as has been found in mitosis.

Beyond these histone tail residues, other residues have been shown to be important for proper meiosis. For example, a histone H4 S1A mutant had ~40% sporulation efficiency relative to wild-type cells [44], albeit a milder deficiency than H3 T3A (4.8% sporulation efficiency relative to wild-type cells). It is also of note that H3 S10A and H3 T3A mutants had different impacts on sporulation efficiency even though both residues are linked to chromosome compaction [24].

We attempted to elucidate how the T3A mutation leads to decreased sporulation efficiency and decreased spore viability. We wondered if the spindle checkpoint is activated in the T3A mutant: The reduced sporulation efficiency could be explained by the SAC delaying meiotic progression. However, sporulation efficiency was not increased in *mad2Δ* T3A double mutants, as would have been predicted. Nevertheless, we also observed that spore viability in *mad2Δ* T3A mutants (24.3%) is much lower than either the T3A mutant (59.3%) or the *mad2Δ* mutant (72.0%) (Figure 5). Therefore, the spindle assembly checkpoint appears to function in suppressing the spore viability deficit seen in T3A cells. Because the spindle assembly checkpoint is activated when there are improper spindle attachments [42] and because the purpose of H3T3 phosphorylation in mitosis is to correct improper kinetochore–microtubule interactions [25], the data suggest that H3T3 phosphorylation in meiosis may promote error correction mechanisms to ensure chromosome biorientation. Interestingly, the spore viability patterns of H3 T3A tetrads and H3 T3A *mad2Δ* double mutant tetrads (Figure 6) were most consistent with homologous chromosome segregation errors in meiosis I. One limitation is that the spore viability assay does not unequivocally distinguish between failed meiosis and defects that could have occurred in the early mitotic divisions after spore germination, which may in some cases have led to an inviable colony. Of note, however, is that Shonn et al. previously observed meiosis I segregation errors in *mad2Δ* cells [58]. Investigating how the SAC in conjunction with H3T3 modulates meiosis and how the spindle assembly checkpoint can rescue the inviability of T3A spores will be interesting future studies.

## 5. Conclusions

Histone H3 threonine 3 has an important role in yeast meiosis, promoting both the efficiency of sporulation and the viability of the resulting haploid cells (spores). Spore viability in cells with a histone H3 T3A amino acid substitution was further reduced when the spindle assembly checkpoint was non-functional. Therefore, the important function of the H3 T3 residue appears to be monitored by the spindle checkpoint.

## Figures and Tables

**Figure 1 biomolecules-15-01202-f001:**
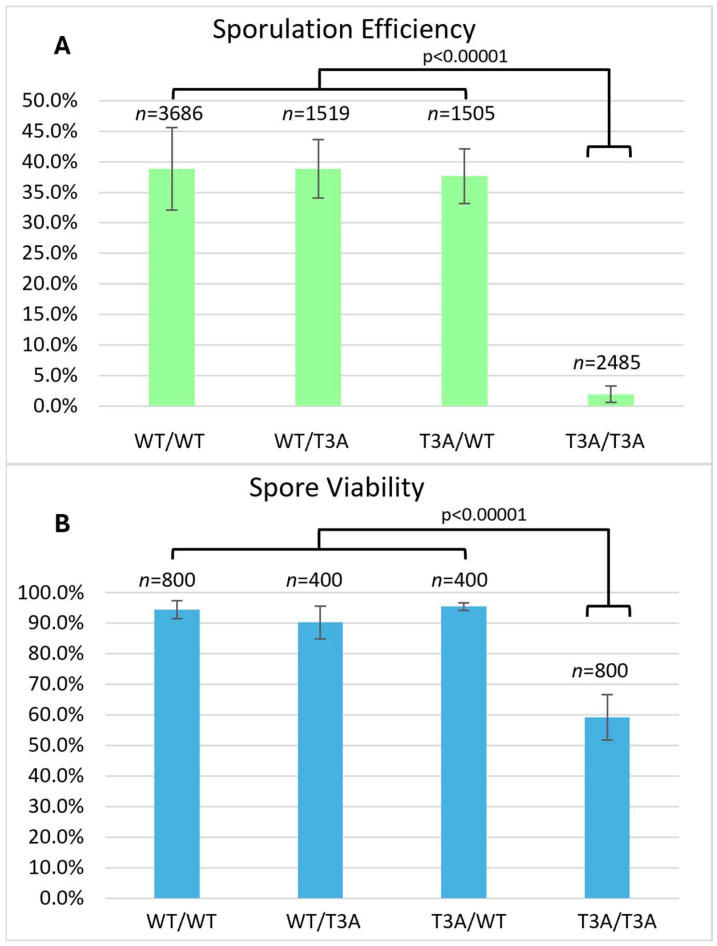
Histone H3 T3A sporulation efficiency and spore viability. (**A**) Sporulation efficiency was calculated as the percentage of cells that had sporulated three days after being plated on sporulation medium. The T3A homozygote (DCY4730) had greatly reduced sporulation efficiency compared to the wild-type (DCY4706) and heterozygotes (DCY4707 and DCY4824). Error bars show standard deviation. (**B**) Spore viability was calculated as the percentage of colonies formed from the tetrads dissected. The T3A homozygote (DCY4730) has reduced viability compared to the wild-type (DCY4706) and heterozygotes (DCY4707 and DCY4824). Error bars show standard deviation.

**Figure 2 biomolecules-15-01202-f002:**
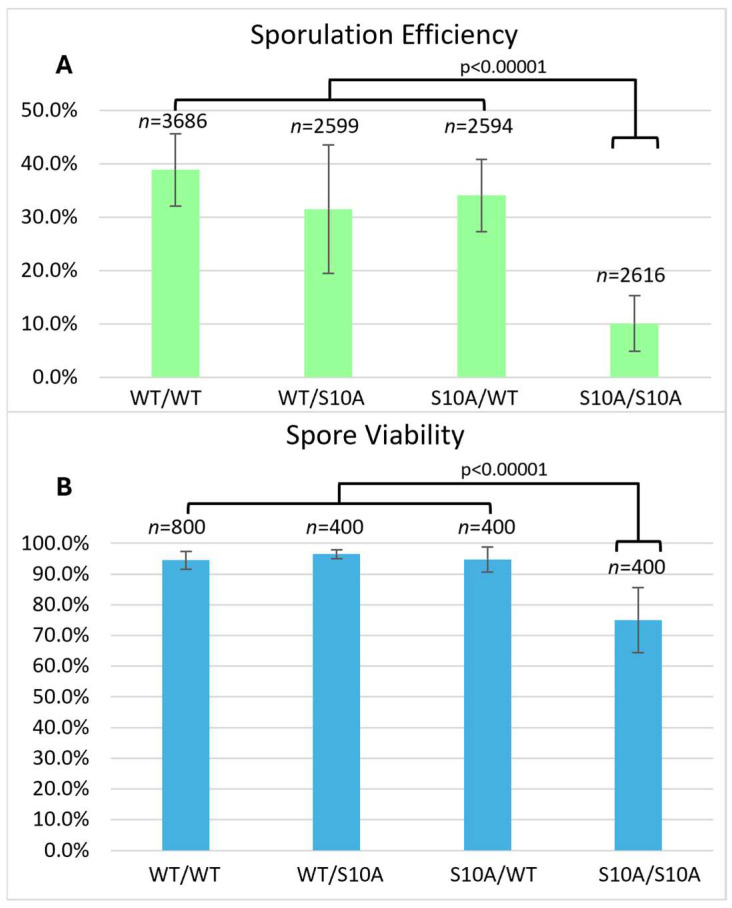
Histone H3 S10A sporulation efficiency and spore viability. (**A**) Sporulation efficiency was calculated as is Figure 1. The S10A homozygote (DCY5047) has reduced sporulation efficiency compared to the wild-type (DCY4706) and heterozygotes (DCY5045 and DCY5043). Interestingly, the two heterozygotes have somewhat reduced sporulation efficiency compared to the wild-type, differences that are statistically significant with *p* < 0.00001 for DCY5045 and *p* = 0.000094 for DCY5043. Error bars show standard deviation. The data for the WT diploid from Figure 1 are shown for comparison. (**B**) Spore viability was calculated as in Figure 1. The S10A homozygote (DCY5047) has reduced viability compared to the wild-type (DCY4706) and heterozygotes (DCY5045 and DCY5043). Error bars show standard deviation. The data for the WT diploid from Figure 1 are shown for comparison.

**Figure 3 biomolecules-15-01202-f003:**
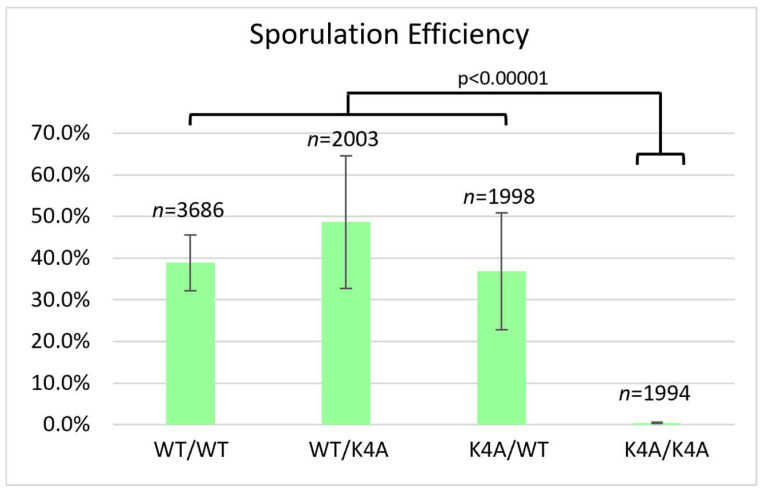
Histone H3 K4A sporulation efficiency. Sporulation efficiency was calculated as in Figure 1. The K4A homozygote (DCY5054) has reduced sporulation efficiency compared to the wild-type (DCY4706) and heterozygotes (DCY5050 and DCY5052). Unexpectedly, one heterozygote had higher and statistically significantly different sporulation efficiency compared to the wild-type, *p* < 0.00001. Error bars show standard deviation. The data for the WT diploid from Figure 1 are shown for comparison.

**Figure 4 biomolecules-15-01202-f004:**
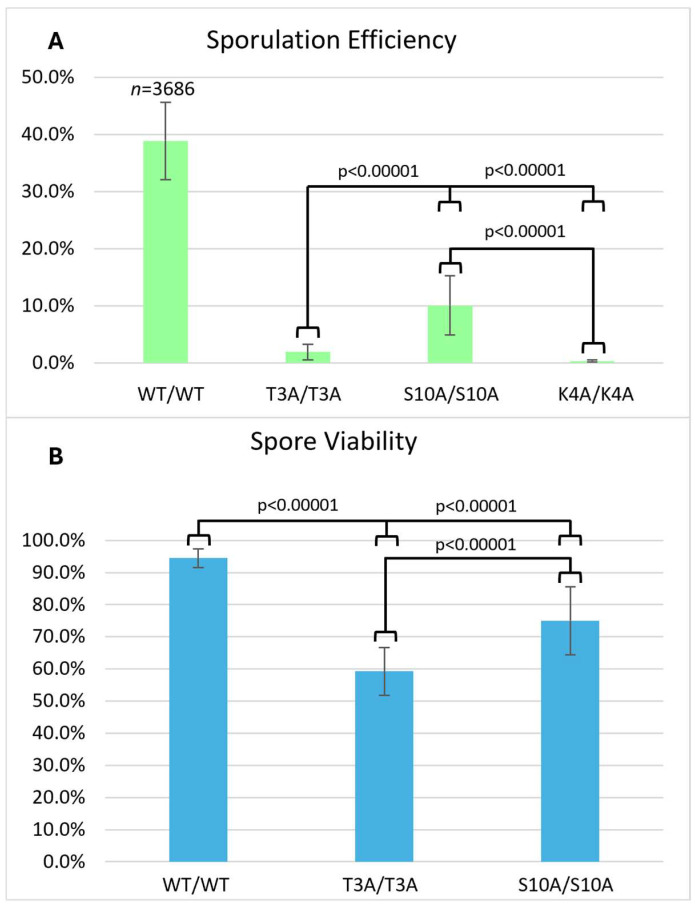
Comparison of sporulation efficiency and spore viability in the histone H3 N-terminal tail mutants. (**A**) Sporulation efficiency was calculated as in Figure 1. Each of the three histone tail mutants tested displayed a severe deficit in sporulation efficiency compared to the wild-type (*p* < 0.00001). Error bars show standard deviation. The data for the WT diploid from Figure 1 are shown for comparison. (**B**) Spore viability was calculated as in Figure 1. Error bars show standard deviation. K4A is not included because its low sporulation efficiency did not permit tetrad analysis. The data for the WT diploid from Figure 1 are shown for comparison.

**Figure 5 biomolecules-15-01202-f005:**
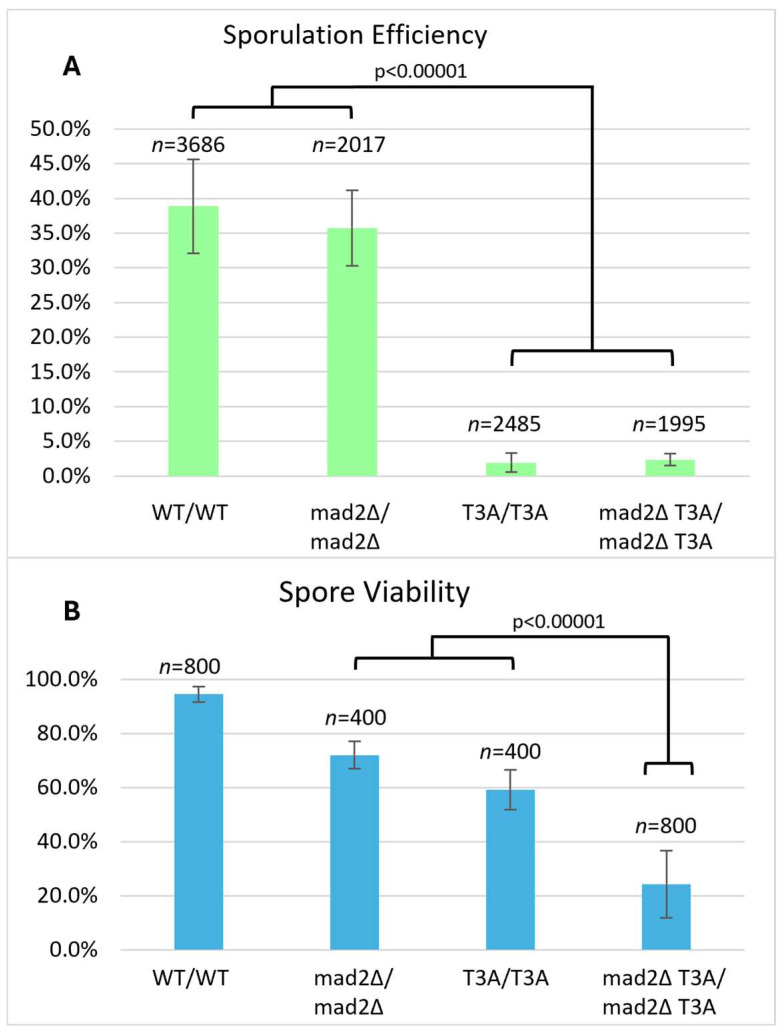
Histone H3 T3A sporulation efficiency and spore viability in combination with *MAD2* deletion. (**A**) Sporulation efficiency was calculated as in Figure 1. The wild-type (DCY4706) and *Δmad2* (DCY5074) mutant have similar sporulation efficiencies (*p* = 0.02 > 0.01). The T3A mutant (DCY4730) and *Δmad2* T3A double mutant (DCY5077) also have similar sporulation efficiencies (*p* > 0.01). Error bars show standard deviation. The data for the WT diploid from Figure 1 are shown for comparison. (**B**) Spore viability was calculated as in Figure 1. All strains shown differ significantly from each other in terms of spore viability (*p* ≤ 0.000015). Notably, the *Δmad2* T3A double mutant (DCY5077) has lower viability than either T3A (DCY4730) or *Δmad2* (DCY5074). Error bars show standard deviation. The data for the WT diploid from Figure 1 are shown for comparison.

**Figure 6 biomolecules-15-01202-f006:**
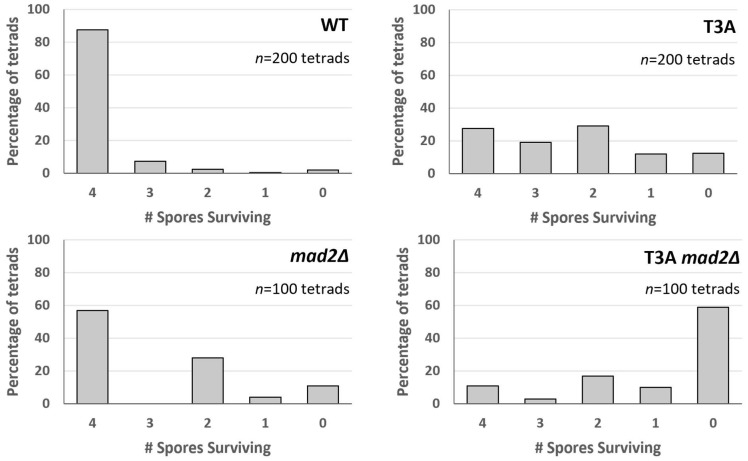
Tetrad spore analysis after meiosis in H3 T3A mutants. Number of spores surviving in each individual tetrad scored after tetrad dissection. The largest increase in the T3A and *mad2Δ* mutants is seen in the 2-spore tetrads. In the T3A *mad2Δ* mutant, there were more 2-spore and 0-spore tetrads than tetrads with 1, 3, or viable 4 spores. This pattern is most consistent with meiosis I chromosome segregation errors. *n* = number of tetrads examined.

**Table 1 biomolecules-15-01202-t001:** Yeast strains used in this study. All yeast strains used in this study are derived from W303. The strain numbers are listed with the mating type (MAT), experimental group, and strain description (genotype).

Number	MAT	Group	Genotype
DCY2459	a	Wild-type	*ade2-1, ura3-1, his3-11,15, trp1-1, leu2-3112, can1-100*
DCY 2460	α	Wild-type	*ade2-1, ura3-1, his3-11,15, trp1-1, leu2-3112, can1-100*
DCY 4595	α	T3A	*hht1-hhf1::hphMX, hht2-T3A-HHF2(URA3)*
DCY 4596	a	T3A	*hht1-hhf1::hphMX, hht2-T3A-HHF2(URA3)*
DCY 4706	a/α	Wild-type	Diploid, 2459 × 2460
DCY 4707	a/α	T3A heterozygote	Diploid, 2459 × 4595
DCY 4730	a/α	T3A homozygote	Diploid, 4595 × 4596
DCY 4824	a/α	T3A heterozygote	Diploid, 2460 × 4596
DCY 4983	a/α	Wild-type	Diploid, 4963 × 4970
DCY 4751	α	K4A	*hht1-hhf1::hphMX, hht2-K4A-HHF2(URA3)*
DCY 4752	a	K4A	*hht1-hhf1::hphMX, hht2-K4A-HHF2(URA3)*
DCY 4753	a	S10A	*hht1-hhf1::hphMX, hht2-S10A-HHF2(URA3)*
DCY 4754	α	S10A	*hht1-hhf1::hphMX, hht2-S10A-HHF2(URA3)*
DCY 5043	a/α	S10A heterozygote	Diploid, 2460 × 4753
DCY 5045	a/α	S10A heterozygote	Diploid, 2459 × 4754
DCY 5047	a/α	S10A homozygote	Diploid, 4753 × 4754
DCY 5050	a/α	K4A heterozygote	Diploid, 2459 × 4751
DCY 5052	a/α	K4A heterozygote	Diploid, 2460 × 4752
DCY 5054	a/α	K4A homozygote	Diploid, 4751 × 4752
DCY 5058	a	*Δmad2*	*mad2::kanMX*
DCY 5061	α	*Δmad2*	*mad2::kanMX*
DCY 5064	α	T3A, *Δmad2*	*hht1-hhf1::hphMX, hht2-T3A-HHF2(URA3), mad2::kanMX*
DCY 5067	a	T3A, *Δmad2*	*hht1-hhf1::hphMX, hht2-T3A-HHF2(URA3), mad2::kanMX*
DCY 5074	a/α	*Δmad2*	Diploid, 5058 × 5061
DCY 5077	a/α	T3A, *Δmad2*	Diploid, 5064 × 5067

**Table 2 biomolecules-15-01202-t002:** Quantification of successful sporulation. Total cells (*n*) attempting sporulation were categorized as having completed sporulation successfully (formation of mature tetrads) versus aberrantly (see Section 2 for full description of the aberrant category).

Yeast Strain	% Successful Sporulation
Wild-type	86% (*n* = 1433)
WT/T3A	80% (*n* = 590) and 78% (*n* = 567)
T3A/T3A	44% (*n* = 48)
*mad2Δ*/*mad2Δ*	64% (*n* = 721)
T3A *mad2Δ*/T3A *mad2Δ*	30% (*n* = 47)
WT/S10A	79% (*n* = 885) and 73% (*n* = 818)
S10A/S10A	37% (*n* = 263)
WT/K4A	77% (*n* = 975) and 79% (*n* = 736)
K4A/K4A	57% (*n* = 7)

## Data Availability

All of the data collected for this study are presented within the figures and tables with the appropriate statistical analyses. The raw data will be made freely and expeditiously available upon request within an Excel file. The data are permanently preserved on a routinely backed-up server at the University of Minnesota.

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
