# Peer review of "Histone H3 N-Terminal Tail Residues Important for Meiosis in Saccharomyces cerevisiae"

_biomolecules, 2025, doi:10.3390/biom15081202_

Round 1

Reviewer 1 Report

Comments and Suggestions for Authors

This manuscript reports the careful analysis of meiosis phenotypes (using sporulation phenotypes as a proxy for meiosis per se) associated with mutating Thr3 in histone H3 to prevent phosphorylation. The experiments are well designed and (for the most part) controlled, and the data are carefully interpreted. I have only a few suggestions for improvement, including a small number of straightforward new experiments. This is considered a "major" revision only because of the request for new experiments.

• “Finally, briefly mention the main aim of the work and highlight the main conclusions.” Currently, the Introduction includes descriptions of the experiments and other details that are unnecessary for this section of the paper. Please condense this section.

• The mutants are made at one locus encoding H3 (HHT2) and the other copy (HHT1) is deleted, including deletion of the adjacent HHF1 gene. Does the slight reduction in histone levels accompanying these deletions contribute to any of the observed phenotypes? Providing T3-mutant HHT1 (and wild-type HHF1) on a plasmid to these cells and looking for a change in phenotype would be an easy way to address this concern.

• Sporulation method: can the authors provide more detail about how the cells were plated on solid sporulation medium? Sporulation on solid medium is convenient and efficient, but it can be difficult to make it reproducible unless a similar density of cells is achieved on the surface of the plate. For example, was a consistent volume of washed overnight YPD culture plated? Also, was sporulation efficiency scored directly on the surface of the plate, or were cells scraped or otherwise transferred in some way to a microscope slide? This is a very small detail but it would help other researchers trying to efficiently replicate the study.

• p. 10 line 267: “Lack of H3T3 phosphorylation therefore activates the SAC in mitosis”. Is there a citation/reference for this statement? What are the data supporting this conclusion?

• Line 365: “To test if this also occurs in meiosis it will be important to determine exactly where H3T3 is phosphorylated in meiotic cells.” What do the authors mean by “where” here? The Thr side chain can only be phosphorylated at one location; do they mean where in the chromatin/genome are H3 molecules phosphorylated on this residue, relative to the centromeres and associated kinetochore attachments? Please clarify the logic here.

• Spore nuclei are visibly smaller due to chromatin compaction (see PMID: 22586276). It would be very easy to see if there are obvious changes in compaction in the nuclei of T3A-mutant spores. This information would contribute to the mechanistic understanding of the defects associated with the mutants.

• Is there any phenotype associated with a phosphomimetic T3D or T3E mutation? I assume these may have been previously tested in mitotic cells. While phosphomimetic mutants can be misleading, in many cases they can be extremely informative, and this study seems like a good context to check.

Author Response

This manuscript reports the careful analysis of meiosis phenotypes (using sporulation phenotypes as a proxy for meiosis per se) associated with mutating Thr3 in histone H3 to prevent phosphorylation. The experiments are well designed and (for the most part) controlled, and the data are carefully interpreted. I have only a few suggestions for improvement, including a small number of straightforward new experiments. This is considered a "major" revision only because of the request for new experiments.

  • “Finally, briefly mention the main aim of the work and highlight the main conclusions.” Currently, the Introduction includes descriptions of the experiments and other details that are unnecessary for this section of the paper. Please condense this section.

Thank you for the recommendation. We have condensed this section of the introduction and removed the extraneous information.

  • The mutants are made at one locus encoding H3 (HHT2) and the other copy (HHT1) is deleted, including deletion of the adjacent HHF1 gene. Does the slight reduction in histone levels accompanying these deletions contribute to any of the observed phenotypes? Providing T3-mutant HHT1 (and wild-type HHF1) on a plasmid to these cells and looking for a change in phenotype would be an easy way to address this concern.

This has been studied before and the findings clearly demonstrated that spore viability after homozygous hht1Δ-hhf1Δ diploids are sporulated is not reduced (Libuda DE, Winston F. Alterations in DNA replication and histone levels promote histone gene amplification in Saccharomyces cerevisiae. Genetics. 2010 Apr;184(4):985-97. doi: 10.1534/genetics.109.113662. Epub 2010 Feb 5. PMID: 20139344; PMCID: PMC2865932). Therefore, the suggested experiment would not provide any new information.

  • Sporulation method: can the authors provide more detail about how the cells were plated on solid sporulation medium? Sporulation on solid medium is convenient and efficient, but it can be difficult to make it reproducible unless a similar density of cells is achieved on the surface of the plate. For example, was a consistent volume of washed overnight YPD culture plated? Also, was sporulation efficiency scored directly on the surface of the plate, or were cells scraped or otherwise transferred in some way to a microscope slide? This is a very small detail but it would help other researchers trying to efficiently replicate the study.

Thank you for the clarifying question. Yes, a consistent volume of washed overnight YPD culture was plated to ensure a similar density. The following description has been added to the methods to improve clarity of this method: “Yeast cells were grown in 5mL liquid YPD medium overnight for 16 hours at 30°C in a rotator. The cells were then centrifuged at 2,000 rpm for 6 minutes to pellet in a tabletop centrifuge, washed three times with ddH2O, and plated on solid sporulation medium in 200μL of ddH2O at room temperature. The cells were grown for three days at room temperature on sporulation medium before sporulation analysis and tetrad dissection.” We also added a statement to the section on microscopy that specifies that the cells were scraped from the plate and transferred to a glass slide. We hope this has cleared up our methods and improved their replicability.

In addition, we would like to point out that three experimental repeats were performed for each condition and that there is good agreement between the repeats. This demonstrated consistency indicates that variation from experiment to experiment, including possible technical inconsistency, was minimal.

  • p. 10 line 267: “Lack of H3T3 phosphorylation therefore activates the SAC in mitosis”. Is there a citation/reference for this statement? What are the data supporting this conclusion?

The reviewer is correct, this statement is not justified, so we have removed it.

  • Line 365: “To test if this also occurs in meiosis it will be important to determine exactly where H3T3 is phosphorylated in meiotic cells.” What do the authors mean by “where” here? The Thr side chain can only be phosphorylated at one location; do they mean where in the chromatin/genome are H3 molecules phosphorylated on this residue, relative to the centromeres and associated kinetochore attachments? Please clarify the logic here.

We have clarified this point. Briefly, as the referee suggests, we predict H3 will be phosphorylated on T3 at centromeres as has been found in mitosis.

  • Spore nuclei are visibly smaller due to chromatin compaction (see PMID: 22586276). It would be very easy to see if there are obvious changes in compaction in the nuclei of T3A-mutant spores. This information would contribute to the mechanistic understanding of the defects associated with the mutants.

On the surface this seems like a reasonable idea. Indeed, meiotic chromosome compaction is affected in mouse meiosis by inhibiting Haspin kinase and we discussed this in our manuscript (see the following citation: Wang Q, Wei H, Du J, Cao Y, Zhang N, Liu X, Liu X, Chen D, Ma W. H3 Thr3 phosphorylation is crucial for meiotic resumption and anaphase onset in oocyte meiosis. Cell Cycle. 2016;15(2):213-24. doi: 10.1080/15384101.2015.1121330. Epub 2015 Dec 4. PMID: 26636626; PMCID: PMC4825905.) However, in yeast it is known that chromosome compaction is very limited during cell division and more specialized methods are required to assess this compaction. Indeed, we have studied this previously using tandem pairs of LacO and TetO sequences integrated at well separated genomic loci on the same chromosome arm, visualized with red and green fluorescent tagged LacI and TetR proteins (for example, see Vas AC, Andrews CA, Kirkland Matesky K, Clarke DJ. In vivo analysis of chromosome condensation in Saccharomyces cerevisiae. Mol Biol Cell. 2007 Feb;18(2):557-68. doi: 10.1091/mbc.e06-05-0454. Epub 2006 Dec 6. PMID: 17151360; PMCID: PMC1783779.) In brief, the proposed experiment would require complex strain construction that likely would take at least several months. We understand the reviewer’s point that this is an interesting experiment, but we consider this to be future work and it would not strengthen the conclusions that we reach in the present manuscript.

  • Is there any phenotype associated with a phosphomimetic T3D or T3E mutation? I assume these may have been previously tested in mitotic cells. While phosphomimetic mutants can be misleading, in many cases they can be extremely informative, and this study seems like a good context to check.

As above, this would require complex additional strain constructions and we consider this future work. In addition, as the reviewer points out phosphomimic mutants can give misleading results and we cannot even make a prediction in this case what the outcome might be.

Reviewer 2 Report

Comments and Suggestions for Authors

The manuscript contains some results interesting not only for yeast geneticists.

It is well written, but it carries some not well supported claims.

The Title "Histone H3 N-terminal tail residues required for meiosis in Saccharomyces cerevisiae" does not reflect the content and should be changed to something like "Histone H3 N-terminal tail residues T3, K4, and S10 are important for meiosis in Saccharomyces cerevisiae strain W303", because it 1) deals only with three not all histone H3 N-terminal tail residues, 2) the mutated residues abrogate meiosis significantly but not entirely and 3) the meiotic effects of at least two of these residues are dependent on genetic background [see ref.27 cited on line 353]. Alternatively to adding "strain W303", data manifesting generality of the mutation effects on S. cerevisiae meiosis [in multiple other strains including wild derivates] must be presented.

Despite repeated claims including in the Abstract, the decreased sporulation efficiency (and thus meiotic effect) of H3T3A has been published [ref.27; what is new for T3A are spore viability number of spores in surviving tetrads, and Mad2 double homozygous mutants]; thus the Abstract lines 12-13 must be rewritten.

The Abstract also claims "Analysis of two other H3 tail mutants, K4A and S10A, revealed different effects on sporulation efficiency and spore viability compared to the T3A mutant, suggesting that these phenotypes may be due to failures in distinct functions." However, the presented data do not exclude the possibility of overlapping function and at least the effect on compound heterozygotes (T3A/S10A and T3A/K4A) should be presented or this claim deleted.

Discussing the intertalk between H3K4 and H3T3 modifications, the authors fail to mention the publications manifesting that binding to H3K4me3 is abrogated by H3T3ph [e.g., of the Spp1 protein important also for yeast meiosis: DOI: 10.1042/BCJ20190091]. These results predicts dependent effects, e.g., compound heterozygote H3 K4A/T3D could have transcription and meiotic DNA break defects. The authors mention that these mutations have mitotic effect; they must also explain, how were their experiments able to distinguish mitotic and meiotic effects in a way comprehensible for general reader.

There are meaningless text repetitions in the Figure legends:

The definitions of Sporulation Efficiency and Spore Viability could appear only once.

The probabilities should be in the images [stars are not needed].

The statistical method should be named and justified in Methods.

The term "test with a significance level of 0.01" should be explained/expanded.

"Error bars show standard deviation.": Do these come from repeating the experiment?

If so, why was not the statistics assessed using t-test?

Bar graphs should then show also the experimental values.

Were the probabilities adjusted for multiple testing? Figs.2-4,5B must show numbers of analyzed cells for WT/WT and legends must state if these cells were overlapping with other experiment data.

The 1st sentence of the Abstract is to be reconsidered and/or modified.

Section 3.1/Fig.1A data should be compared to T3A data from ref.27.

L298: What do you mean by "product of spore viabilities"? "The predicted viability"?

L299-300: a statistical test should be presented supporting lower spore viability.

L303-322: statistical tests supporting the claims "The largest increases in the T3A mad2Δ mutant are seen in the 2 spore and 0 spore tetrads." are missing.

L390: Future experiments should also assess the detailed phenotypes of H3 T3D and S10D including double homozygotes and compound heterozygotes.

Round 2

Reviewer 1 Report

Comments and Suggestions for Authors

I am satisfied with the authors' response to my suggestions and associated revisions to the manuscript.

Author Response

Thank you for helping us improve our manuscript. We appreciate your constructive comments and expertise.

Reviewer 2 Report

Comments and Suggestions for Authors

The authors solved or explained most of the issues I raised in my review except for a few exceptions.

What I meant by commenting "a statistical test should be presented supporting lower spore viability" is Chi-square test comparing the viability of the double mutants
and viability expected from the percentage counted (42.7%- on line 270) by multiplying the percentages of single mutants: 97 of 400 versus 171 of 400 :
R code: chisq.test(matrix(c(97,303,171,229),nrow=2,ncol=2)) 
this test produces p-value supporting the conclusion that the difference between these percentages is not due to chance (sufficient number of spores analyzed).

A similar tests should be run for the claims based on data from Fig.6.

The legend of Fig.6 and text could benefit from unifying the terminology, as it is describing "inviable" spores in the text versus "surviving" spores in Fig.6.

L293: “or viable 4 spores.” do you mean “or 4 viable spores.” ?

I do not see any change or explanation for my comment ". The authors mention that these mutations have mitotic effect; they must also explain, how were their experiments able to distinguish mitotic and meiotic effects in a way comprehensible for general reader."

Comments on the Quality of English Language

Some word order could be improved.

Author Response

Thank you for helping us improve our manuscript. We appreciate your constructive comments and expertise.

What I meant by commenting "a statistical test should be presented supporting lower spore viability" is Chi-square test comparing the viability of the double mutants
and viability expected from the percentage counted (42.7%- on line 270) by multiplying the percentages of single mutants: 97 of 400 versus 171 of 400 :
R code: chisq.test(matrix(c(97,303,171,229),nrow=2,ncol=2)) 
this test produces p-value supporting the conclusion that the difference between these percentages is not due to chance (sufficient number of spores analyzed).

Thanks. We have done this statistical analysis in R as you recommended and added this information to the text.

A similar tests should be run for the claims based on data from Fig.6.

Good suggestion! We have done similar tests comparing viable spores to the wild-type for the section of the results that goes over Figure 6 and included the statistical information in the text.

The legend of Fig.6 and text could benefit from unifying the terminology, as it is describing "inviable" spores in the text versus "surviving" spores in Fig.6.

Thank you for the suggestion. We have changed the phrasing in the text to match the figures for clarity.

L293: “or viable 4 spores.” do you mean “or 4 viable spores.” ?

Thanks for catching this mistake. We meant 4 viable spores, and we have edited the text to fix this typo.

I do not see any change or explanation for my comment ". The authors mention that these mutations have mitotic effect; they must also explain, how were their experiments able to distinguish mitotic and meiotic effects in a way comprehensible for general reader."

Sorry for missing this point earlier. We agree it is not possible to distinguish meiosis errors from mitotic errors in the spore viability assay because it relies on the germinated spores growing into colonies through mitotic divisions. Still, the strains are viable as haploids and homozygous diploids, so there cannot be a severe loss of viability during vegetative growth. Nevertheless, we included an explanation on this point in the revised manuscript as follows, “One limitation is that the spore viability assay does not unequivocally distinguish between failed meiosis and defects that could have occurred in the early mitotic divisions after spore germination, that may in some cases have led to an inviable colony.”